

# Forecasting Bitcoin closing price series using linear regression and neural networks models

Nicola Uras*, Lodovica Marchesi*, Michele Marchesi and Roberto Tonelli

Department of Mathematics and Computer Science, University of Cagliari, Cagliari, Italy
* These authors contributed equally to this work.

## ABSTRACT

In this article we forecast daily closing price series of Bitcoin, Litecoin and Ethereum cryptocurrencies, using data on prices and volumes of prior days. Cryptocurrencies price behaviour is still largely unexplored, presenting new opportunities for researchers and economists to highlight similarities and differences with standard financial prices. We compared our results with various benchmarks: one recent work on Bitcoin prices forecasting that follows different approaches, a well-known paper that uses Intel, National Bank shares and Microsoft daily NASDAQ closing prices spanning a 3-year interval and another, more recent paper which gives quantitative results on stock market index predictions. We followed different approaches in parallel, implementing both statistical techniques and machine learning algorithms: the Simple Linear Regression (SLR) model for uni-variate series forecast using only closing prices, and the Multiple Linear Regression (MLR) model for multivariate series using both price and volume data. We used two artificial neural networks as well: Multilayer Perceptron (MLP) and Long short-term memory (LSTM). While the entire time series resulted to be indistinguishable from a random walk, the partitioning of datasets into shorter sequences, representing different price "regimes", allows to obtain precise forecast as evaluated in terms of Mean Absolute Percentage Error(MAPE) and relative Root Mean Square Error (relativeRMSE). In this case the best results are obtained using more than one previous price, thus confirming the existence of time regimes different from random walks. Our models perform well also in terms of time complexity, and provide overall results better than those obtained in the benchmark studies, improving the state-of-the-art.

## INTRODUCTION

Bitcoin is the world's most valuable cryptocurrency, a form of electronic cash, invented by an unknown person or group of people using the pseudonym Satoshi Nakamoto (*Nakamoto, 2008*), whose network of nodes was started in 2009. Although the system was introduced in 2009, its actual use began to grow only from 2013. Therefore, Bitcoin is a new entry in currency markets, though it is officially considered as a commodity rather than a currency, and its price behaviour is still largely unexplored, presenting new

Corresponding authors
Nicola Uras, nicola.uras@unica.it
Lodovica Marchesi,
lodovica.marchesi@unica.it

opportunities for researchers and economists to highlight similarities and differences with standard financial currencies, also in view of its very different nature with respect to more traditional currencies or commodities. The price volatility of Bitcoin is far greater than that of fiat currencies (*Briere, Oosterlinck & Szafarz, 2013*), providing significant potential in comparison to mature financial markets (*McIntyre & Harjes, 2014*; *Cocco, Tonelli & Marchesi, 2019a*; *Cocco, Tonelli & Marchesi, 2019b*). According to CoinMarketCap (https://www.coinmarketcap.com), one of the most popular sites that provides almost real-time data on the listing of the various cryptocurrencies in global exchanges, on May 2019 Bitcoin market capitalization value is valued at approximately 105 billion of USD. Hence, forecasting Bitcoin price has also great implications both for investors and traders. Even if the number of bitcoin price forecasting studies is increasing, it still remains limited (*Mallqui & Fernandes, 2018*). In this work, we approach the forecast of daily closing price series of the Bitcoin cryptocurrency using data on prices and volumes of prior days. We compare our results with three well-known recent papers, one dealing with Bitcoin prices forecasting using other approaches, one forecasting Intel, National Bank shares and Microsoft daily NASDAQ prices and one on stock market index forecasting using fusion of machine learning techniques.

The first paper we compare to, tries to predict three of the most challenging stock market time series data from NASDAQ historical quotes, namely Intel, National Bank shares and Microsoft daily closed (last) stock price, using a model based on chaotic mapping, firefly algorithm, and Support Vector Regression (SVR) (*Kazem et al., 2013*). In the second one *Mallqui & Fernandes (2018)* used different machine learning techniques such as Artificial Neural Networks (ANN) and Support Vector Machines (SVM) to predict, among other things, closing prices of Bitcoin. The third paper we consider in our work proposes a two stage fusion approach to forecast stock market index. The first stage involves SVR. The second stage uses ANN, Random Forest (RF) and SVR (*Patel et al., 2015*). We decided to predict these three share prices to give a sense of how Bitcoin is different from traditional markets. Moreover, to enrich our work, we applied the models also to two other two well-know cryptocurrencies: Ethereum and Litecoin. In this work we forecast daily closing price series of Bitcoin cryptocurrency using data of prior days following different approaches in parallel, implementing both statistical techniques and machine learning algorithms. We tested the chosen algorithms on two datasets: the first consisting only of the closing prices of the previous days; the second adding the volume data. Since Bitcoin exchanges are open 24/7, the closing price reported on *Coinmarketcap* we used, refers to the price at 11:59 PM UTC of any given day. The implemented algorithms are Simple Linear Regression (SLR) model for univariate series forecast, using only closing prices; a Multiple Linear Regression (MLR) model for multivariate series, using both price and volume data; a Multilayer Perceptron and a Long Short-Term Memory neural networks tested using both the datasets. The first step consisted in a statistical analysis of the overall series. From this analysis we show that the entire series are not distinguishable from a random walk. If the series were truly random walks, it would not be possible to make any forecasts. Since we are interested in prices and not in price variations, we avoided the time series differencing technique by introducing and using the novel presented approach. Therefore, each time

series was segmented in shorter overlapping sequences in order to find shorter time regimes that do not resemble a random walk so that they can be easily modeled. Afterwards, we run all the algorithms again on the partitioned dataset.

The reminder of this article is organized as follows. 'Literature Review' presents the methodology, briefly describing the data, their pre-processing, and finally the models used. 'Methods' presents and discuss the results. 'Results' concludes the article.

## LITERATURE REVIEW

Over the years many algorithms have been developed for forecasting time series in stock markets. The most widely adopted are based on the analysis of past market movements (*Agrawal, Chourasia & Mittra, 2013*). Among the others, *Armano, Marchesi & Murru (2015)* proposed a prediction system using a combination of genetic and neural approaches, having as inputs technical analysis factors that are combined with daily prices. *Enke & Mehdiyev (2013)* discussed a hybrid prediction model that combines differential evolution-based fuzzy clustering with a fuzzy inference neural network for performing an index level forecast. *Kazem et al. (2013)* presented a forecasting model based on chaotic mapping, firefly algorithm, and support vector regression (SVR) to predict stock market prices. Unlike other widely studied time series, still very few researches have focused on bitcoin price prediction. In a recent exploration *McNally, Roche & Caton (2018)* tried to ascertain with what accuracy the direction of Bitcoin price in USD can be predicted using machine learning algorithms like LSTM (Long short-term memory) and RNN (Recurrent Neural Network). *Naimy & Hayek (2018)* tried to forecast the volatility of the Bitcoin/USD exchange rate using GARCH (Generalized AutoRegressive Conditional Heteroscedasticity) models. *Sutiksno et al. (2018)* studied and applied $\alpha$-Sutte indicator and Arima (Autoregressive Integrated Moving Average) methods to forecast historical data of Bitcoin. *Stocchi & Marchesi (2018)* proposed the use of Fast Wavelet Transform to forecast Bitcoin prices. *Yang & Kim (2016)* examined a few complexity measures of the Bitcoin transaction flow networks, and modeled the joint dynamic relationship between these complexity measures and Bitcoin market variables such as return and volatility. *Bakar & Rosbi (2017)* presented a forecasting Bitcoin exchange rate model in high volatility environment, using autoregressive integrated moving average (ARIMA) algorithms. *Catania, Grassi & Ravazzolo (2018)* studied the predictability of cryptocurrencies time series, comparing several alternative univariate and multivariate models in point and density forecasting of four of the most capitalized series: Bitcoin, Litecoin, Ripple and Ethereum, using univariate Dynamic Linear Models and several multivariate Vector Autoregressive models with different forms of time variation. *Vo & Xu (2017)* used knowledge of statistics for financial time series and machine learning to fit the parametric distribution and model and forecast the volatility of Bitcoin returns, and analyze its correlation to other financial market indicators. Other approaches try to predict stock market index using fusion of machine learning techniques (*Patel et al., 2015*). *Akcora et al. (2018)* introduced a novel concept of chainlets, or bitcoin subgraphs, to evaluate the local topological structure of the Bitcoin graph over time and the role of chainlets on bitcoin price formation and dynamics.

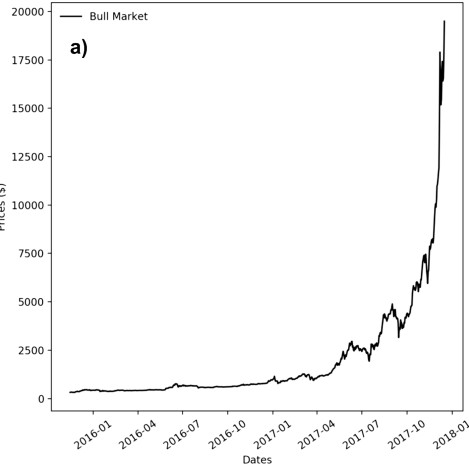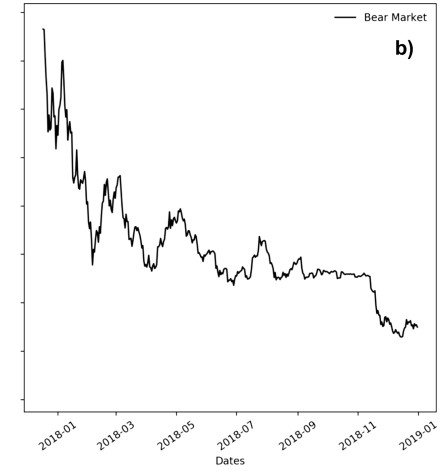

**Figure 1 Bull (A) and Bear (B) price dynamics for Bitcoin market.**

*Greave & Au (2015)* predicted the future price of bitcoin investigating the predictive power of blockchain network-based, in particular using the bitcoin transaction graph. Since the cryptocurrencies market is at an early stage, the cited papers that deals with forecasting bitcoin prices had the opportunity to train and test their models on a quite narrow dataset. In particular, bitcoin market has been at first characterized by an almost constantly ascending price trend, the so-called bull-market condition. However, since 2018, it has been characterized by a strong descending price trend, the so-called bear-market condition. Therefore, the cited papers trained their models on data of the first market condition, and tested them on data of the second type. These market conditions are shown in Fig. 1 (Fig. 1A: bull-market condition; Fig. 1B: bear-market condition). Our study spans over a period of more than 4 years, characterized by different price dynamics. Therefore, we were able to train and test our models, including in each stage both bull- and bear- market conditions. For these reasons, our study enriches the state-of-the-art, as it is the most updated and deals with the biggest and more complete dataset.

## METHODS

In this section we first introduce some notions on time series analysis, which helped us to take the operational decisions about the algorithms we used and to better understand the results presented in the following. Then, we present the dataset we used, including its pre-processing analysis. Finally we introduce our proposed algorithms with the metrics employed to evaluate their performance and the statistical tools we adopted.

### Time series analysis
#### Time series components
Any time series is supposed to consist of three systematic components that can be described and modelled. These are 'base level', 'trend' and 'seasonality', plus one non-systematic component called 'noise'. The base level is defined as the average value in the series. A trend is observed when there is an increasing or decreasing slope in the time series.

Seasonality is observed when there is a repeated pattern between regular intervals, due to seasonal factors. Noise represents the random variations in the series. Every time series is a combination of these four components, where base level and noise always occur, whereas trend and seasonality are optional. Depending on the nature of the trend and seasonality, a time series can be described as an additive or multiplicative model. This means that each observation in the series can be expressed as either a sum or a product of the components (*Hyndman & Athanasopoulos, 2014*). An additive model is described by following the linear equation:

$$y(t) = BaseLevel + Trend + Seasonality + Noise \qquad (1)$$

A multiplicative model is instead represented by the following non-linear equation:

$$y(t) = BaseLevel * Trend * Seasonality * Noise \qquad (2)$$

An additive model would be used when the variations around the trend does not vary with the level of the time series whereas a multiplicative model would be appropriate if the trend is proportional to the level of the time series. This method of time series decomposition is called "classical decomposition" (*Hyndman & Athanasopoulos, 2014*).

### *Statistical measures*

The statistical measures we calculated for each time series are the mean, labelled with $\mu$, the standard deviation $\sigma$ and the trimmed mean $\bar{\mu}$, obtained discarding a portion of data from both tails of the distribution. The trimmed mean is less sensitive to outliers than the mean, but it still gives a reasonable estimate of central tendency and can be very helpful for time series with high volatility.

### Collected data

We tested our algorithms on six daily price series. Three of them are stock market series, all the data were extracted from the 'Historical Data' available on (http://www.finance.yahoo.com) website; the other ones are cryptocurrencies, namely Bitcoin, Ethereum and Litecoin price daily series, all the data were extracted from (http://www.coinmarketcap.com) website.

- Daily stock market prices for Microsoft Corporation (MSFT), from 9/12/2007 to 11/11/2011.
- Daily stock market prices for Intel Corporation (INTC), from 9/12/2007 to 11/11/2010.
- Daily stock market prices for National Bankshares Inc. (NKSH), from 6/27/2008 to 8/29/2011.
- Daily Bitcoin, Ethereum and Litecoin price series, from 15/11/2015 to 12/03/2020.

We state once more that we choose these price series and the related time intervals as benchmark to compare our results with well known literature results obtained by using other methods.

Specifically, we have chosen for the stock market series the same time intervals chosen in *Kazem et al. (2013)*. The choice of Bitcoin as criptocurrency is quite natural since it represents about 60 % of the Total Market Capitalization. We chose Ethereum and

Litecoin since they are among the most important and well-known cryptocurrencies. It is worth noting that, for the stock market series we used the same data of the work we compare to, whereas for the cryptocurrencies we used all the available data to have more significant results.

The dataset was divided into two sets, a training part and a testing part. After some empirical test the partition of the data which lead us to optimal solutions was 80 % of the daily data for the training dataset and the remaining for the testing dataset.

## Data pre-processing

For both models we prepared our dataset in order to have a set of inputs ($X$) and outputs ($Y$) with temporal dependence. We performed a one-step ahead forecast: our output $Y$ is the value from the next (future) point of time while the inputs $X$ are one or several values from the past, i.e., the so called *lagged* values. From now on we identify the number of used lagged values with the *lag* parameter. In the Linear Regression and *Univariate* LSTM models the dataset includes only the daily closing price series, hence there is only one single *lag* parameter for the *close* feature. On the contrary, in the Multiple Linear Regression and *Multivariate* LSTM models the dataset includes both *close* and *volume (USD)* series, hence we use two different *lag* parameters, one for the *close* and one for the *volume* feature. In both cases, we attempted to optimize the predictive performance of the models by varying the *lag* from 1 to 10.

## Univariate versus multivariate forecasting

A univariate forecast consists of predicting time series made by observations belonging to a single feature recorded over time, in our case the closing price of the series considered. A multivariate forecast is a forecast in which the dataset consists of the observations of several features. In our case we used:

- for BTC, ETH and LTC series all the features provided by *Coinmarketcap* website: Open, High, Low, Close, Volume.
- for MSFT, INTC, NKSH series all the features provided by *Yahoofinance* website: Date, Open, High, Low, Close, Volume.

We observed that adding features to the dataset did not lead to better predictions, but performance and sometimes also results worsened. For this reason, we decided to use in the multivariate analysis only the *close* and *volume* features, that provided the best results.

## Statistical analysis

As a first step we carried out a statistical analysis in order to check for non-stationarity in the time series. We used the *augmented Dickey-Fuller test* and *autocorrelation plots* (*Banerjee et al., 1993*; *Box & Jenkins, 1976*). A stochastic process with a *unit root* is non-stationary, namely shows statistical properties that change over time, including mean, variance and covariance, and can cause problems in predictability of time series models. A common process with *unit root* is the *random walk*. Often price time series show some characteristics which makes them indistinguishable from a random walk. The presence of such a process can be tested using a *unit root* test.

The *ADF* test is a statistical test that can be used to test for a *unit root* in a univariate process, such as time series samples. The null hypothesis $H_0$ of the *ADF* test is that there is a *unit root*, with the alternative $H_a$ that there is no *unit root*. The most significant results provided by this test are the *observed test statistic*, the Mackinnon's approximate $p$-value and the *critical values* at the 1%, 5% and 10% levels.

The test statistic is simply the value provided by the *ADF* test for a given time series. Once this value is computed it can be compared to the relevant critical value for the Dickey-Fuller Test.

Critical values, usually referred to as $\alpha$ levels, are an error rate defined in the hypothesis test. They give the probability to reject the null hypothesis $H_0$. So if the observed test statistic is less than the critical value (keep in mind that ADF statistic values are always negative (*Banerjee et al., 1993*), then the null hypothesis $H_0$ is rejected and no *unit root* is present.

The $p$-value is instead the probability to get a "more extreme" test statistic than the one observed, based on the assumed statistical hypothesis $H_0$, and its mathematical definition is shown in Eq. (3).

$$p_{value} = P\left(t \geq t_{observed} \,\middle|\, H_0\right). \tag{3}$$

The $p$-value is sometimes called *significance*, actually meaning the closeness of the $p$-value to zero: the lower the $p$-value, the higher the significance.

In our analysis we performed this test using the *adfuller()* function provided by the *statsmodels* Python library, and we chose a *significance level* of 5%.

Furthermore, the *autocorrelation plot*, also known as *correlogram*, allowed us to calculate the correlation between each observation and the observations at previous time steps, called *lag values*. In our case we employed the *autocorrelation_plot()* function provided by the python *Pandas* library (*Mckinney, 2011*).

## Forecasting

We decided to follow two different approaches: the first uses two well-known statistical methods: Linear Regression (LR) and Multiple Linear Regression (MLR). The second uses two very common neural networks (NN): Multilayer Perceptron (MLP) NN and Long Short-Term Memory (LSTM) NN. The reasons of this choices are explained below.

### Linear regression and multiple linear regression

Linear regression is a linear approach for modelling the relationship between a dependent variable and one independent variable, represented by the main equation:

$$y = b_0 + \vec{b}_1 \cdot \vec{x}_1, \tag{4}$$

where $y$ and $\vec{x}_1$ are the dependent and the independent variable respectively, while $b_0$ is the intercept and $\vec{b}_1$ is the vector of slope coefficients. In our case the components of the vector $\vec{x}_1$, our independent variable, are the values of the closing prices of the previous days. Therefore, $\vec{x}_1$ size is the value of the *lag* parameter. In our case $y$ represents the closing price to be predicted.

This algorithm aims to find the curve that best fits the data, which best describes the relation between the dependent and independent variable. The algorithm finds the best fitting line plotting all the possible trend lines through our data and for each of them calculates and stores the amount $(y - \bar{y})^2$, and then choose the one that minimizes the squared differences sum $\sum_i (y_i - \bar{y}_i)^2$, namely the line that minimizes the distance between the real points and those crossed by the line of best fit.

We then tried to forecast with multiple independent variables, adding to the *close* price feature the observations of several features, including *volume*, *highest value* and *lowest value* of the previous day. These information were gained from *Coinmarketcap* website. In these cases we used a Multiple Linear Regression model (MLR). The MLR equation is:

$$y = b_0 + \vec{b}_1 \cdot \vec{x}_1 + ... + \vec{b}_n \cdot \vec{x}_n = b_0 + \sum_{i=1}^{n} \vec{b}_i \cdot \vec{x}_i \tag{5}$$

where the index *i* refers to a particular independent variable and *n* is the dimension of the independent variables space.

We used the Linear and Multiple regression model of *scikit learn* (*Pedregosa et al., 2012*). We decided to use this two models for several reasons: they are simple to write, use and understand, they are fast to compute, they are commonly used models and fit well to datasets with few features, like ours. Their disadvantage is that they can model only linear relationships.

### Multilayer perceptron

A multilayer perceptron (MLP) is a feedforward artificial neural network that generates a set of outputs from a set of inputs. It consists of at least three layers of neurons: an input layer, a hidden layer and an output layer. Each neuron, apart from the input ones, has a nonlinear activation function. MLP uses backpropagation for training the network. In our model we keep the structure as simple as possible, with a single hidden layer. Our inputs are the closing prices of the previous days, where the number of values considered depends on the *lag* parameter. The output is the forecast price. The optimal number of neurons were found by optimizing the network architecture on the number of neurons itself, varying it in an interval between 5 and 100. We used the Python *Keras* library (*Chollet, 2015*).

### LSTM networks

Long Short-Term Memory networks are nothing more than a prominent variations of Recurrent Neural Network (RNN). RNN's are a class of artificial neural network with a specific architecture oriented at recognizing patterns in sequences of data of various kinds: texts, genomes, handwriting, the spoken word, or numerical time series data emanating from sensors, markets or other sources (*Hochreiter & Schmidhuber, 1997*). Simple recurrent neural networks are proven to perform well only for short-term memory and are unable to capture long-term dependencies in a sequence. On the contrary, LSTM networks are a special kind of RNN, able at learning long-term dependencies. The model is organized in cells which include several operations. LSTM hold an internal state variable, which is passed from one cell to another and modified by Operation Gates (forget gate, input gate, output gate). These gates control how much of the internal state is passed to the output

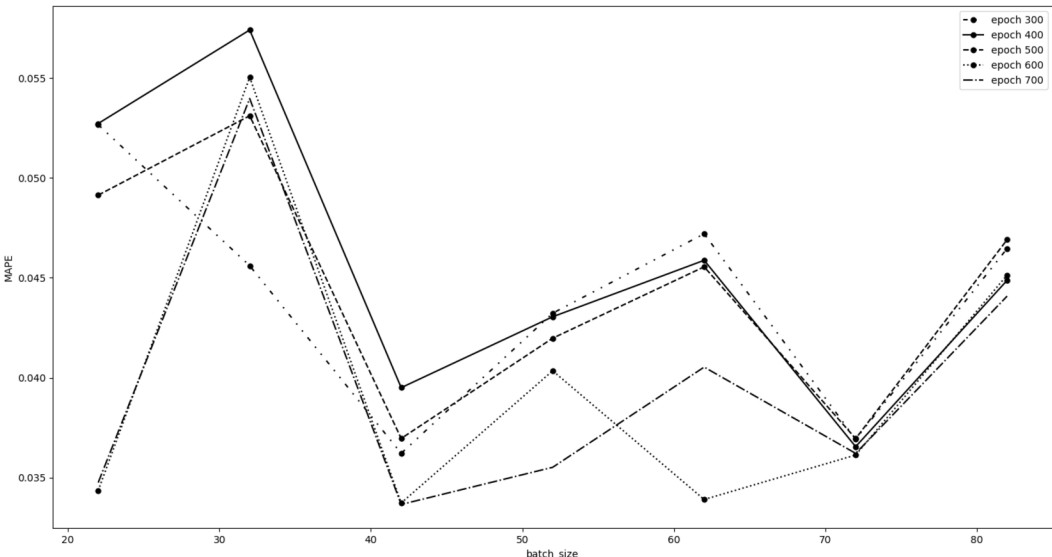

**Figure 2**  **Bitcoin hyperparameters tuning results.**

and work in a similar way to other gates. These three gates have independent weights and biases, hence the network will learn how much of the past output and of the current input to retain and how much of the internal state to send out to the output.

In our case the inputs are the closing prices of the previous days and the number of values considered depends on the *lag* parameter. The output is the forecast price. We used the Keras framework for deep learning. Our model consists of one stacked LSTM layer with 64 units each and the densely connected output layer with one neuron. We used Adam optimizer and MSE (mean squared error) as a loss.

We optimized our *LSTM* model searching for the best set of *epochs* and *batch size* "hyperparameters" values. These hyperparameters strongly depend on the number of observations available for the experiment. Due to the recently birth of the cryptocurrency markets, the dimensions of our datasets are quite limited (around 1000 observations), therefore we decided to vary the *epochs* hyperparameter from 300 to 800 with a step of 100. The Keras LSTM algorithm we used sets as default value for *batch size* 32. So, for each fixed *epoch*, we trained the model varying the *batch size* within the interval $[22, 82]$ with a step of 10. We did not take into account values less than 300 epochs, nor greater than 800 in order to avoid *underfitting* and *overfitting* problems. Furthermore, we did not consider *batch size* values less than 22, since they would lead to extremely long training times. Similarly, *batch size* values greater than 82 would not allow to find a good local minimum point of the chosen loss function during the learning procedure. The results obtained during the hyperparameters tuning are shown in Fig. 2.

This figure shows the *MAPE* error as a function of the *batch size* hyperparameter, for each fixed epoch. As can be seen from the figure, we considered the *batch size* equal to 72 to be the optimal value. In fact, it is an excellent compromise, having a low MAPE value, which is also practically the same for all tested *epochs*. The optimal choice for the *epochs*

hyperparameter is 600, which is the one that minimizes the MAPE error for *batch size* equal to 72, and is consistently among the best choices for almost all batch sizes considered. Therefore, the best set of *epochs* and *batch size* "hyperparameters" values we chose is 600 and 72, respectively.

## Time regimes

The time series considered are found to be indistinguishable from a random walk. This peculiarity is common for time series of financial markets, and in our case is confirmed by the predictions of the models, in which the best result is obtained considering only the price of the previous day.

The purpose is to find an approach that allow us to avoid time series differencing technique, in view of the fact that we are interested in prices and not in price variations represented by integrated series of $d$-order. For this reason, each time series was segmented into short partially overlapping sequences, in order to find if shorter time regimes are present, where the series do not resemble a random walk. Finally, to continue with the forecasting procedure, a train and a test set were identified within each time regime.

For each regime we always sampled 200 observations - namely 200 daily prices. The beginning of the next regime is obtained with a shift of 120 points from the previous one. Thus, every regime is 200 points wide and has 80 points in common with the following one.

We chose a regime length of 200 days because, in this way, we obtain at least 5 regimes (from 5 to 12) for each time series to test the effectiveness of our algorithms, without excessively reducing the number of samples needed for training and testing. The choice was determined also according to the following: we performed the augmented Dickey-Fuller test on subsets of the data, starting from the whole set and progressively reducing the data window and sliding it through the data. The first subset of data that does not behave as random walks appears at time interval of 230 days, which we rounded to 200.

Since the time series considered have different lengths, the partition in regimes has generated:

- Bitcoin, Ethereum and Litecoin: 12 regimes
- Microsoft: 8 regimes
- Intel and National Bankshares: 5 regimes

From a mathematical point of view, the used approach can be described as follows.

Let us target a vector $\overrightarrow{OA}$ along the $t$ axis, with length 200. This vector is identified by the points $O(1,0)$, $A(a,0) \equiv (200,0)$. The length of this vector represents the width of each time regime.

Let $\overrightarrow{OH}$ be a fixed translation vector along the $t$ axis, identified by the points $O(1,0)$ and $H(h,0) \equiv (120,0)$. The length of $\overrightarrow{OH}$ represents the translation size.

For the sake of simplicity, let us label the $\overrightarrow{OA}$ and $\overrightarrow{OH}$ vectors with $\vec{A}$ and $\vec{H}$.

Let $\vec{A}'$ be the vector $\vec{A}$ shifted by $\vec{H}$ and $\vec{A}^n$ the vector $\vec{A}$ shifted by $n$ times $\vec{H}$.

Therefore, the vector that identifies the $n$th sequence to be sampled along the series is given by:

$$\vec{A}^n = \vec{A} + n\vec{H} \tag{6}$$

where $n \in \left[0, \frac{D-A}{h}\right]$, being $D$ the dimension of the sampling space, $A$ the time regimes width and $h$ the translation size.

So the $n$th time regime is given by:

$$R^n = f\left(\vec{A}^n\right) = f\left(\vec{A} + n\vec{H}\right) \tag{7}$$

where $f$ is the function that maps the values along the $t$ axis (dates) to the respective regimes $y$ values (actual prices).

## Performance Measures

To evaluate the effectiveness of different approaches, we used the *relative* Root Mean Square Error (rRMSE) and the Mean Absolute Percentage Error (MAPE), defined respectively as:

$$relativeRMSE = \sqrt{\frac{1}{N}\sum_{i=1}^{N}\left(\frac{y_i - f_i}{y_i}\right)^2} \tag{8}$$

$$MAPE = \frac{1}{N}\sum_{i=1}^{N}\left|\frac{y_i - f_i}{y_i}\right| \tag{9}$$

In both formulas $y_i$ and $f_i$ represent the actual and forecast values, and $N$ is the number of forecasting periods. These are scale free performance measures, so that they are well appropriate to compare model performance results across series with different orders of magnitude, as in our study.

## RESULTS

### Time series analysis

In Fig. 3 we report the decomposition of Bitcoin (Fig. 3A-3D) and Microsoft (Fig 3E-3H) time series, for comparison purposes, as obtained using the *seasonal_decompose()* method, provided by the Python *statsmodels* library (*Skipper & Perktold., 2010*).

The *seasonal_decompose()* method requires to specify whether the model is additive or multiplicative. In the Bitcoin time series, the trend of increase at the beginning is almost absent (from around 2016-04 to 2017-02); in later years, the frequency and the amplitude of the cycle appears to change over time. The Microsoft time series shows a non-linear seasonality over the whole period, with frequency and amplitude of the cycles changing over time. These considerations suggest that the model is multiplicative. Furthermore, if we look at the residuals, they look quite random, in agreement with their definitions. The Bitcoin residuals are likewise meaningful, showing periods of high variability in the later years of the series.

It is also possible to group the data at seasonal intervals, observing how the values are distributed and how they evolve over time. In our work we grouped the data of the same

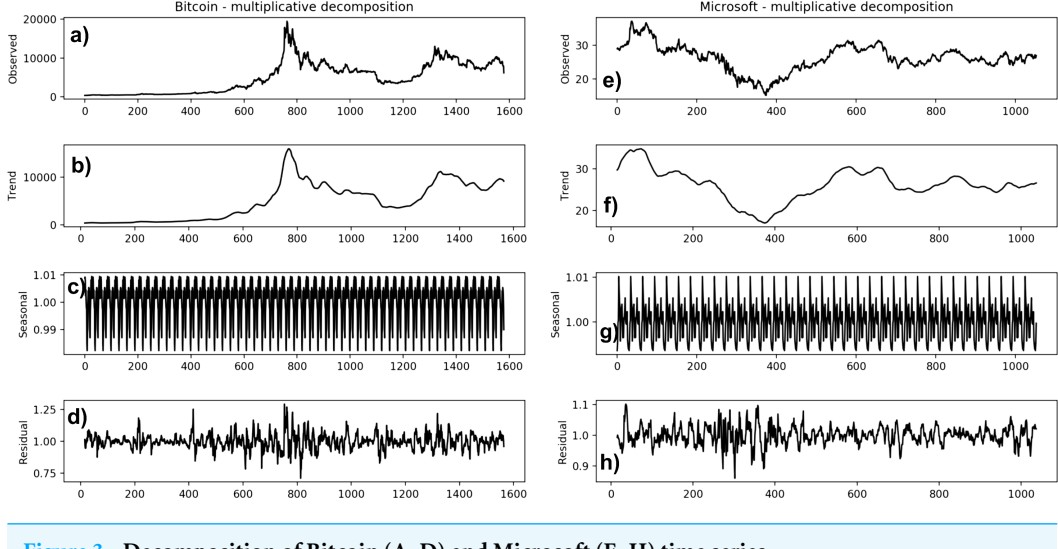

**Figure 3  Decomposition of Bitcoin (A–D) and Microsoft (E–H) time series.**

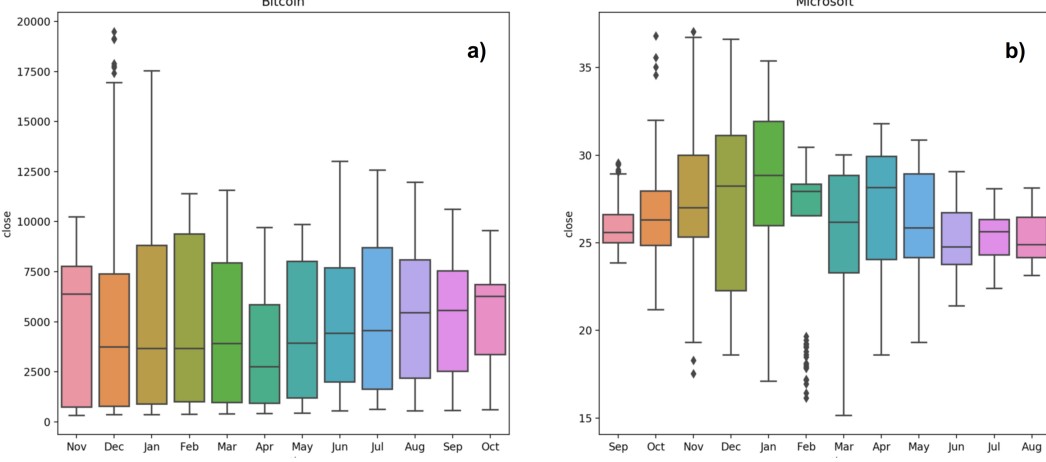

**Figure 4  Seasonality of Bitcoin (A) and Microsoft (B) time series.**

month over the years we considered. This is achieved with the 'Box plot' of month-wide distribution, shown in Fig. 4 (Fig. 4A: Bitcoin; Fig. 4B: Microsoft). The Box plot is a standardized way of displaying the distribution of data based on five numbers summary: minimum, first quartile, median, third quartile and maximum. The box of the plot is a rectangle which encloses the middle half of the sample, with an end at each quartile. The length of the box is thus the inter-quartile range of the sample. The other dimension of the box has no meaning. A line is drawn across the box at the sample median. Whiskers sprout from the two ends of the box defining the outliers range. The box length gives an indication of the sample variability, and for the Bitcoin samples shows a large variance, in almost all months, except for April, September and October. Not surprisingly, bitcoin volatility is much higher than Microsoft one. The line crossing the box shows where the

**Table 1  Time series statistical measures.**

| Series | μ | σ | $\bar{\mu}$ |
|--------|-----|-----|-----|
| BTC | 4,931,3 | 3,970,0 | 4,593,1 |
| ETH | 216,8 | 239,8 | 171,2 |
| LTC | 55,9 | 58,0 | 45,6 |
| MSFT | 26,2 | 3,9 | 26,3 |
| INTC | 19,9 | 3,6 | 19,9 |
| NKSH | 24,3 | 3,9 | 24,5 |

sample is centred, i.e., the median. The position of the box in its whiskers and the position of the line in the box also tell us whether the sample is symmetric or skewed, either to the right or to the left. The plot shows that the Bitcoin monthly samples are therefore skewed to the right. The top whiskers is much longer than the bottom whiskers and the median is gravitating towards the bottom of the box. This is due to the very high prices that Bitcoin reached throughout the period between 2017 and 2018. These large values tend to skew the sample statistics. In Microsoft, an alternation between samples skewed to the left and samples skewed to the right occurs, except for the sample of October that shows a symmetric distribution. Lack of symmetry entails one tail being longer than the other, distinguishing between heavy-tailed or light-tailed populations. In the Bitcoin case we can state that the majority of the samples are left skewed populations with short tails. Microsoft shows an alternation between heavy-tailed and light-tailed distributions. We can see that some Microsoft samples, particularly those with long tails, present outliers, representing anomalous values. This is due to the fact that heavy tailed distributions tend to have many outliers with very high values. The heavier the tail, the larger the probability that you will get one or more disproportionate values in a sample.

Tables 1 and 2 show the statistics calculated for each time series and for each short time regime. The unit of measurement of the values in the tables is the US dollar ($). In Table 1 we can observe that the only series for which the trimmed mean, obtained with *trim_mean()* method provided by the Python *scipy* library (*Jones, Oliphant & Peterson, 2001*), with a cut-off percentage of 10%, is significantly different from the mean are Bitcoin, Ethereum and Litecoin. In particular the trimmed mean decreased. This is due to the fact that these cryptocurrencies, for a long period of time, registered a large price increment and this implies a shift of the mean to the right (i.e., to highest prices). This confirms that cryptocurrencies distribution is right-skewed. Table 2 shows that stock market series time regimes present a lower $\sigma$ than BTC, ETH and LTC ones, namely that cryptocurrencies distribution has higher variance.

Figures 5 and 6 show the autocorrelation plots of BTC and MSFT series. The others stock market series are not presented because they show the same features of the MSFT series. Both autocorrelation plots (Figs. 5C and 6C) show a strong autocorrelation between the current price and the closest previous observations and a linear fall-off from there to the first few hundred lag values. We then tried to make the series stationary by taking the *first difference*. The autocorrelation plots of the 'differences series' (Figs. 5C and 5D) show

**Table 2  Regimes statistical measures.**

| Series | h | μ | σ | μ̄ |
|---|---|---|---|---|
| BTC | 0 | 419,7 | 39,6 | 421,6 |
| | 120 | 551,2 | 97,3 | 549,6 |
| | 240 | 707,9 | 122,5 | 693,2 |
| | 360 | 1110,1 | 358,8 | 1048,8 |
| | 480 | 2481,2 | 1107,4 | 2414,0 |
| | 600 | 7446,4 | 4808,8 | 6870,7 |
| | 720 | 10359,6 | 3082,8 | 9966,1 |
| | 840 | 7536,5 | 1130,1 | 7424,8 |
| | 960 | 5810,9 | 1382,3 | 5859,4 |
| | 1,080 | 4509,6 | 1101,3 | 4349,9 |
| | 1,200 | 8016,9 | 2752,9 | 8048,3 |
| | 1,320 | 9154,5 | 1477,4 | 9080,2 |
| ETH | 0 | 6,0 | 4,6 | 5,8 |
| | 120 | 11,7 | 2,0 | 11,6 |
| | 240 | 10,8 | 1,7 | 10,8 |
| | 360 | 34,6 | 39,0 | 26,3 |
| | 480 | 195,8 | 114,6 | 194,5 |
| | 600 | 441,9 | 281,8 | 385,5 |
| | 720 | 695,9 | 251,4 | 682,0 |
| | 840 | 487,4 | 159,1 | 486,4 |
| | 960 | 239,6 | 118,0 | 228,2 |
| | 1,080 | 144,6 | 34,0 | 141,8 |
| | 1,200 | 204,7 | 52,5 | 201,1 |
| | 1,320 | 186,8 | 42,5 | 181,7 |
| LTC | 0 | 3,5 | 0,4 | 3,4 |
| | 120 | 3,9 | 0,5 | 3,9 |
| | 240 | 3,9 | 0,2 | 3,9 |
| | 360 | 8,2 | 8,1 | 6,2 |
| | 480 | 33,8 | 19,3 | 33,3 |
| | 600 | 102,6 | 85,4 | 86,1 |
| | 720 | 167,0 | 65,0 | 163,7 |
| | 840 | 107,6 | 40,2 | 105,3 |
| | 960 | 52,9 | 17,9 | 52,2 |
| | 1080 | 50,5 | 19,9 | 48,7 |
| | 1200 | 87,4 | 23,8 | 85,7 |
| | 1320 | 67,2 | 22,3 | 64,5 |

no significant relationship between the lagged observations. All correlations are small, close to zero and below the 95% and 99% confidence levels.

As regards the *augmented Dickey-Fuller* results, shown in Table 3, looking at the observed *test statistics*, we can state that all the series follows a unit root process. We remind that the null hypothesis $H_0$ of the *ADF* test is that there is a *unit root*. In particular, all the observed

**Table 2** (*continued*)

| Series | h | μ | σ | μ̄ |
|---|---|---|---|---|
| MSFT | 0 | 30,7 | 2,8 | 30,5 |
| | 120 | 26,1 | 3,2 | 26,4 |
| | 240 | 20,6 | 3,9 | 20,4 |
| | 360 | 22,8 | 3,8 | 22,8 |
| | 480 | 28,2 | 2,3 | 28,4 |
| | 600 | 26,8 | 2,2 | 26,7 |
| | 720 | 26,1 | 1,3 | 26,1 |
| | 840 | 26,0 | 1,2 | 26,0 |
| INTC | 0 | 23,5 | 2,4 | 23,5 |
| | 120 | 20,0 | 3,6 | 20,3 |
| | 240 | 15,4 | 2,3 | 15,1 |
| | 360 | 17,3 | 2,3 | 17,4 |
| | 480 | 20,6 | 1,4 | 20,4 |
| NKSH | 0 | 18,5 | 0,9 | 18,5 |
| | 120 | 22,2 | 3,0 | 22,2 |
| | 240 | 26,5 | 1,4 | 26,5 |
| | 360 | 25,9 | 1,9 | 26,0 |
| | 480 | 26,5 | 2,5 | 26,3 |

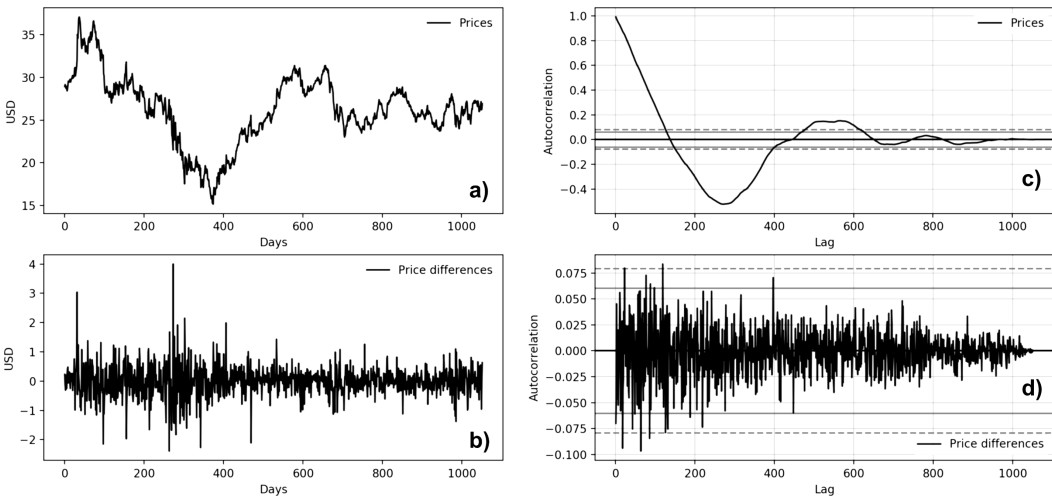

**Figure 5** **Microsoft time series autocorrelation plots (A: Microsoft price behavior; B: first-difference prices plot; C: prices autocorrelation plot; D: price differences autocorrelation plot).**

*test statistics* are greater than those associated to all significance levels. This implies that we can not reject the null hypothesis $H_0$, but does not imply that the null hypothesis is true.

Observing the *p-values*, we notice that for the stock market series we have a low probability to get a "more extreme" test statistic than the one observed under the null hypothesis $H_0$. Precisely, for both *MSFT* and *INTC* we got a probability of 29%, for *NKSH* a probability of 25%. The same considerations also apply to the Bitcoin, Ethereum and

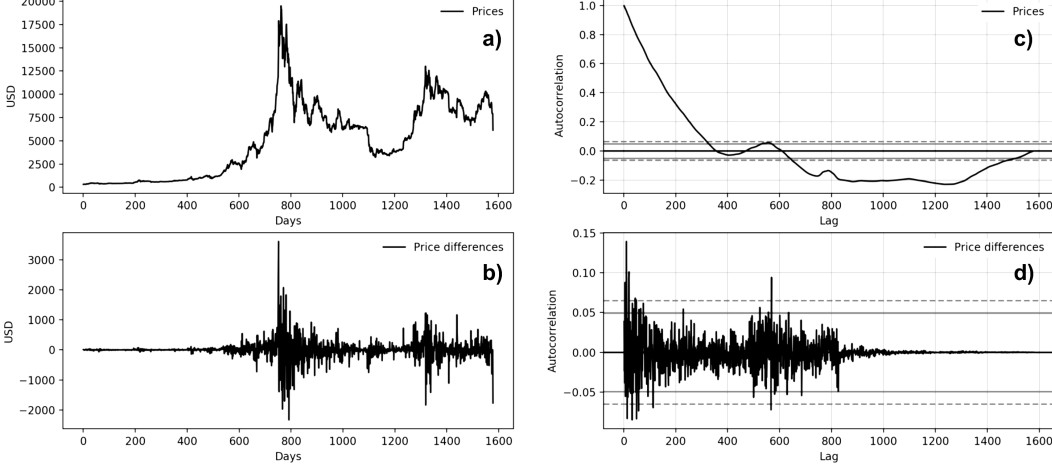

**Figure 6** Bitcoin time series autocorrelation plots (A: Bitcoin price behavior; B: first-difference prices plot; C: prices autocorrelation plot; D: price differences autocorrelation plot).

**Table 3** Augmented Dickey-Fuller test results.

| Series | ADF statistic | p-value |
|--------|---------------|---------|
| BTC | −2.12 | 0.24 |
| ETH | −2.17 | 0.22 |
| LTC | −2.34 | 0.16 |
| MSFT | −1.98 | 0.29 |
| INTC | −1.98 | 0.29 |
| NKSH | −2.10 | 0.25 |

Litecoin cryptocurrency time series. We conclude that $H_0$ can not be rejected and so each time series present a *unit root* process.

We conclude that all the considered series show the statistical characteristics typical of a *random walk*.

## Time series forecasting

Tables 4 and 5 show the best results, in terms of MAPE and rRMSE, obtained with the different algorithms applied to the entire series. From now on, let us label the closing and the volume features *lag* parameters with $k_p$ and $k_v$ respectively. In particular, Table 4 reports the results obtained using the *Linear Regression* algorithm for univariate series forecast, using only closing prices, and the *Multiple Linear Regression* model for multivariate series, using both price and volume data.

Table 5 shows the results obtained with the *LSTM* neural network, distinguishing between *univariate LSTM*, using only closing prices, and *multivariate LSTM*, using both price and volume data.

Small values of the *MAPE* and *rRMSE* evaluation metrics suggest accurate predictions and good performance of the considered model.

**Table 4 Linear and multiple linear regression results.**

| | Linear regression | | | Multiple linear regression | | | |
|---|---|---|---|---|---|---|---|
| Series | MAPE | rRMSE | $k_p$ | MAPE | rRMSE | $k_p$ | $k_v$ |
| BTC | 0.026 | 0.040 | 1 | 0.026 | 0.037 | 1 | 1 |
| ETH | 0.031 | 0.049 | 1 | 0.039 | 0.053 | 6 | 3 |
| LTC | 0.034 | 0.050 | 1 | 0.045 | 0.058 | 2 | 2 |
| MSFT | 0.011 | 0.015 | 1 | 0.011 | 0.015 | 1 | 1 |
| INTC | 0.013 | 0.017 | 1 | 0.013 | 0.017 | 1 | 1 |
| NKSH | 0.014 | 0.019 | 12 | 0.013 | 0.018 | 7 | 5 |

**Table 5 Univariate and multivariate LSTM results.**

| | Univariate LSTM | | | Multivariate LSTM | | | |
|---|---|---|---|---|---|---|---|
| Series | MAPE | rRMSE | $k_p$ | MAPE | rRMSE | $k_p$ | $k_v$ |
| BTC | 0.027 | 0.041 | 1 | 0.038 | 0.048 | 2 | 1 |
| ETH | 0.034 | 0.052 | 6 | 0.057 | 0.076 | 2 | 1 |
| LTC | 0.035 | 0.051 | 1 | 0.039 | 0.054 | 1 | 1 |
| MSFT | 0.012 | 0.015 | 1 | 0.012 | 0.015 | 1 | 2 |
| INTC | 0.013 | 0.017 | 2 | 0.013 | 0.017 | 1 | 1 |
| NKSH | 0.014 | 0.020 | 7 | 0.013 | 0.018 | 1 | 2 |

From the analysis of the series in their totality, it appears that linear models outperforms neural networks. However, for both models, the majority of best results are obtained for a *lag* of 1,thus confirming our hypothesis that the series are indistinguishable from a random walk.

In order to perform the time series forecasting, we also implemented a *Multi-Layer Perceptron* model. Since the *LSTM* network outperforms the *MLP* one, we decided to show only the *LSTM* results. This is probably due to the particular architecture of the LSTM network, that is able to capture long-term dependencies in a sequence.

It should be noted that better predictions are obtained for stock market series rather than for the cryptocurrencies one. In particular, the best result is obtained for Microsoft series, with a MAPE of 0, 011 and $k_p$ equal to 1. This is probably due to the high price fluctuations that Bitcoin and the other cryptocurrencies have suffered during the investigated time interval. This is confirmed by the statistics shown in Table 1. It must be noted that the addition of the *volume* feature to the dataset does not improve the predictions.

In order to perform prices forecast we changed the approach and decided to split the time series analysis using shorter time windows of 200 points, shifting the windows by 120 points, with the aim of finding local time regimes where the series do not follow the global random walk pattern.

Tables 6 and 7 show the results obtained with our approach of partitioning the series into shorter sequences. Let us label the moving step forward with *h*. Particularly, in Table 6 are presented the results obtained using the *Linear Regression* algorithm for univariate series forecast, using only closing prices, and the *Multiple Linear Regression* model for

**Table 6** LR and MLR results with time regimes.

| Series | h | Linear regression | | | Multiple Linear Regression | | | |
| | | MAPE | rRMSE | $k_p$ | MAPE | rRMSE | $k_p$ | $k_v$ |
|---|---|---|---|---|---|---|---|---|
| BTC | 0 | 0.015 | 0.025 | 4 | 0.012 | 0.014 | 8 | 10 |
| | 120 | 0.007 | 0.010 | 7 | 0.007 | 0.011 | 1 | 1 |
| | 240 | 0.029 | 0.050 | 4 | 0.031 | 0.052 | 5 | 1 |
| | 360 | 0.034 | 0.041 | 1 | 0.037 | 0.045 | 1 | 2 |
| | 480 | 0.041 | 0.062 | 2 | 0.039 | 0.061 | 2 | 1 |
| | 600 | 0.065 | 0.082 | 2 | 0.065 | 0.080 | 2 | 2 |
| | 720 | 0.028 | 0.035 | 1 | 0.026 | 0.035 | 1 | 5 |
| | 840 | 0.017 | 0.024 | 7 | 0.018 | 0.024 | 7 | 1 |
| | 960 | 0.030 | 0.040 | 4 | 0.029 | 0.040 | 1 | 10 |
| | 1.080 | 0.029 | 0.039 | 1 | 0.022 | 0.031 | 3 | 3 |
| | 1.200 | 0.018 | 0.025 | 8 | 0.021 | 0.026 | 8 | 2 |
| | 1.320 | 0.020 | 0.026 | 5 | 0.021 | 0.027 | 7 | 7 |
| ETH | 0 | 0.045 | 0.060 | 7 | 0.042 | 0.056 | 10 | 6 |
| | 120 | 0.022 | 0.029 | 1 | 0.022 | 0.028 | 1 | 1 |
| | 240 | 0.031 | 0.047 | 4 | 0.033 | 0.046 | 1 | 3 |
| | 360 | 0.053 | 0.078 | 1 | 0.053 | 0.078 | 2 | 2 |
| | 480 | 0.048 | 0.077 | 1 | 0.050 | 0.077 | 1 | 1 |
| | 600 | 0.060 | 0.080 | 1 | 0.053 | 0.069 | 3 | 8 |
| | 720 | 0.039 | 0.051 | 1 | 0.036 | 0.049 | 1 | 7 |
| | 840 | 0.048 | 0.070 | 7 | 0.064 | 0.084 | 5 | 1 |
| | 960 | 0.051 | 0.068 | 1 | 0.055 | 0.071 | 4 | 1 |
| | 1.080 | 0.032 | 0.046 | 3 | 0.020 | 0.027 | 10 | 7 |
| | 1.200 | 0.024 | 0.031 | 8 | 0.022 | 0.029 | 1 | 8 |
| | 1.320 | 0.025 | 0.033 | 1 | 0.028 | 0.035 | 1 | 1 |
| LTC | 0 | 0.027 | 0.034 | 4 | 0.023 | 0.027 | 8 | 8 |
| | 120 | 0.011 | 0.018 | 3 | 0.011 | 0.017 | 1 | 4 |
| | 240 | 0.030 | 0.046 | 5 | 0.031 | 0.047 | 5 | 2 |
| | 360 | 0.075 | 0.098 | 1 | 0.074 | 0.094 | 3 | 3 |
| | 480 | 0.073 | 0.111 | 1 | 0.074 | 0.112 | 1 | 1 |
| | 600 | 0.077 | 0.096 | 2 | 0.058 | 0.074 | 8 | 7 |
| | 720 | 0.040 | 0.049 | 1 | 0.040 | 0.047 | 1 | 1 |
| | 840 | 0.032 | 0.045 | 9 | 0.031 | 0.043 | 9 | 3 |
| | 960 | 0.047 | 0.060 | 3 | 0.048 | 0.062 | 1 | 1 |
| | 1.080 | 0.037 | 0.047 | 9 | 0.023 | 0.028 | 7 | 7 |
| | 1.200 | 0.026 | 0.032 | 8 | 0.027 | 0.034 | 8 | 1 |
| | 1.320 | 0.026 | 0.036 | 1 | 0.026 | 0.037 | 1 | 1 |

**Table 6** (*continued*)

| Series | h | Linear regression | | | Multiple Linear Regression | | | |
| | | MAPE | rRMSE | $k_p$ | MAPE | rRMSE | $k_p$ | $k_v$ |
|---|---|---|---|---|---|---|---|---|
| MSFT | 0 | 0.015 | 0.018 | 1 | 0.015 | 0.017 | 1 | 3 |
| | 120 | 0.037 | 0.045 | 6 | 0.035 | 0.044 | 6 | 4 |
| | 240 | 0.015 | 0.019 | 7 | 0.015 | 0.019 | 9 | 6 |
| | 360 | 0.010 | 0.014 | 3 | 0.012 | 0.018 | 1 | 1 |
| | 480 | 0.011 | 0.015 | 2 | 0.010 | 0.012 | 3 | 7 |
| | 600 | 0.009 | 0.011 | 4 | 0.009 | 0.011 | 5 | 1 |
| | 720 | 0.008 | 0.011 | 7 | 0.007 | 0.009 | 10 | 8 |
| | 840 | 0.012 | 0.015 | 1 | 0.012 | 0.015 | 1 | 10 |
| INTC | 0 | 0.014 | 0.019 | 5 | 0.013 | 0.017 | 6 | 10 |
| | 120 | 0.036 | 0.045 | 7 | 0.035 | 0.043 | 7 | 4 |
| | 240 | 0.017 | 0.022 | 5 | 0.017 | 0.022 | 2 | 3 |
| | 360 | 0.012 | 0.015 | 1 | 0.012 | 0.015 | 1 | 1 |
| | 480 | 0.016 | 0.020 | 1 | 0.016 | 0.020 | 3 | 5 |
| NKSH | 0 | 0.019 | 0.023 | 8 | 0.019 | 0.023 | 9 | 6 |
| | 120 | 0.014 | 0.018 | 9 | 0.013 | 0.017 | 10 | 4 |
| | 240 | 0.014 | 0.018 | 4 | 0.012 | 0.016 | 1 | 4 |
| | 360 | 0.019 | 0.026 | 2 | 0.019 | 0.026 | 2 | 1 |
| | 480 | 0.009 | 0.012 | 7 | 0.009 | 0.012 | 10 | 5 |

multivariate series, using both price and volume data. This approach, has the advantage of being simple to implement and requires low computational complexity. Nevertheless, has led to good results, similar to those present in the literature, if not better as in the Microsoft, Bitcoin and National Bankshares cases, where the MAPE error is lower that 1%.

Table 7 shows the results obtained with the *LSTM* neural network, distinguishing between *univariate LSTM*, using only closing prices, and *multivariate LSTM*, using both price and volume data. For each time regimes we show the best results obtained on a specific time window defined by the $k_p$ and $k_v$ values reported in Tables 6 and 7. Note that we highlighted the best results in bold. In particular, it is worth noting that introducing the time regimes, the best result is obtained for the Bitcoin time series, outperforming also the financial ones.

These results show how such innovative partitioning approach allowed us to avoid the "random walk problem", finding that best results are obtained using more than one previous price. Furthermore, this method leads to a significant improvement in predictions. It is worth noting that, from this analysis the best result arise from the Bitcoin series, with a *MAPE* error of 0.007, a temporal window $k_p$ of 7 and a translation step $h$ of 120, obtained using both regression models and LSTM network.

Another interesting consideration that arises from the results is that, as stated previously in the analysis of the series in their entirety, the linear regression models generally outperform the neural networks ones, while in the short-time regimes approach the different models yielded to similar results.

**Table 7  Univariate and multivariate LSTM results with time regimes.**

| Series | h | Univariate LSTM | | | Multivariate LSTM | | | |
|---|---|---|---|---|---|---|---|---|
| | | MAPE | rRMSE | $k_p$ | MAPE | rRMSE | $k_p$ | $k_v$ |
| BTC | 0 | 0.022 | 0.034 | 3 | 0.021 | 0.030 | 3 | 1 |
| | 120 | 0.007 | 0.011 | 4 | 0.007 | 0.010 | 2 | 1 |
| | 240 | 0.044 | 0.058 | 3 | 0.065 | 0.077 | 3 | 1 |
| | 360 | 0.088 | 0.105 | 2 | 0.187 | 0.233 | 3 | 3 |
| | 480 | 0.043 | 0.066 | 4 | 0.041 | 0.061 | 1 | 1 |
| | 600 | 0.068 | 0.088 | 1 | 0.078 | 0.127 | 2 | 1 |
| | 720 | 0.027 | 0.035 | 2 | 0.027 | 0.043 | 1 | 2 |
| | 840 | 0.017 | 0.023 | 1 | 0.017 | 0.031 | 3 | 1 |
| | 960 | 0.027 | 0.035 | 6 | 0.033 | 0.067 | 2 | 1 |
| | 1.080 | 0.025 | 0.038 | 3 | 0.030 | 0.106 | 3 | 1 |
| | 1.200 | 0.021 | 0.028 | 1 | 0.024 | 0.033 | 1 | 1 |
| | 1.320 | 0.018 | 0.025 | 1 | 0.020 | 0.028 | 1 | 2 |
| ETH | 0 | 0.051 | 0.065 | 6 | 0.054 | 0.068 | 3 | 1 |
| | 120 | 0.022 | 0.028 | 1 | 0.023 | 0.031 | 1 | 3 |
| | 240 | 0.034 | 0.049 | 1 | 0.035 | 0.048 | 1 | 2 |
| | 360 | 0.217 | 0.248 | 5 | 0.284 | 0.349 | 3 | 3 |
| | 480 | 0.049 | 0.077 | 2 | 0.050 | 0.076 | 1 | 1 |
| | 600 | 0.074 | 0.109 | 3 | 0.164 | 0.396 | 1 | 1 |
| | 720 | 0.039 | 0.052 | 3 | 0.037 | 0.079 | 3 | 1 |
| | 840 | 0.067 | 0.092 | 1 | 0.052 | 0.252 | 1 | 1 |
| | 960 | 0.053 | 0.067 | 1 | 0.062 | 0.101 | 1 | 1 |
| | 1.080 | 0.031 | 0.042 | 3 | 0.039 | 0.082 | 1 | 1 |
| | 1.200 | 0.026 | 0.035 | 1 | 0.025 | 0.049 | 1 | 3 |
| | 1.320 | 0.021 | 0.031 | 2 | 0.022 | 0.031 | 1 | 1 |
| LTC | 0 | 0.045 | 0.054 | 5 | 0.063 | 0.079 | 3 | 1 |
| | 120 | 0.010 | 0.016 | 2 | 0.011 | 0.018 | 3 | 1 |
| | 240 | 0.035 | 0.052 | 6 | 0.051 | 0.069 | 1 | 1 |
| | 360 | 0.395 | 0.409 | 6 | 0.397 | 0.443 | 3 | 2 |
| | 480 | 0.086 | 0.117 | 3 | 0.090 | 0.120 | 3 | 1 |
| | 600 | 0.136 | 0.164 | 1 | 0.167 | 0.431 | 1 | 3 |
| | 720 | 0.040 | 0.051 | 3 | 0.040 | 0.075 | 1 | 2 |
| | 840 | 0.034 | 0.045 | 1 | 0.035 | 0.062 | 1 | 2 |
| | 960 | 0.047 | 0.059 | 1 | 0.053 | 0.107 | 2 | 1 |
| | 1.080 | 0.047 | 0.055 | 1 | 0.034 | 0.121 | 1 | 3 |
| | 1.200 | 0.026 | 0.035 | 1 | 0.026 | 0.048 | 1 | 3 |
| | 1.320 | 0.028 | 0.038 | 2 | 0.028 | 0.038 | 1 | 1 |

| Series | h | Univariate LSTM | | | Multivariate LSTM | | | |
|---|---|---|---|---|---|---|---|---|
| | | MAPE | rRMSE | $k_p$ | MAPE | rRMSE | $k_p$ | $k_v$ |
| MSFT | 0 | 0.014 | 0.017 | 1 | 0.014 | 0.017 | 1 | 2 |
| | 120 | 0.121 | 0.139 | 1 | 0.054 | 0.064 | 3 | 1 |
| | 240 | 0.017 | 0.023 | 2 | 0.017 | 0.023 | 1 | 3 |
| | 360 | 0.017 | 0.021 | 4 | 0.031 | 0.044 | 3 | 1 |
| | 480 | 0.012 | 0.015 | 1 | 0.012 | 0.016 | 1 | 2 |
| | 600 | 0.009 | 0.012 | 3 | 0.009 | 0.012 | 3 | 1 |
| | 720 | 0.008 | 0.011 | 4 | 0.010 | 0.014 | 2 | 1 |
| | 840 | 0.012 | 0.016 | 4 | 0.012 | 0.016 | 3 | 1 |
| INTC | 0 | 0.015 | 0.019 | 1 | 0.014 | 0.018 | 1 | 1 |
| | 120 | 0.056 | 0.068 | 1 | 0.069 | 0.091 | 3 | 3 |
| | 240 | 0.017 | 0.021 | 3 | 0.017 | 0.022 | 3 | 1 |
| | 360 | 0.012 | 0.015 | 1 | 0.013 | 0.017 | 1 | 1 |
| | 480 | 0.017 | 0.021 | 1 | 0.020 | 0.025 | 1 | 1 |
| NKSH | 0 | 0.021 | 0.027 | 1 | 0.023 | 0.027 | 3 | 1 |
| | 120 | 0.015 | 0.018 | 6 | 0.014 | 0.019 | 1 | 3 |
| | 240 | 0.016 | 0.022 | 1 | 0.017 | 0.022 | 1 | 3 |
| | 360 | 0.020 | 0.027 | 1 | 0.023 | 0.030 | 1 | 3 |
| | 480 | 0.010 | 0.014 | 1 | 0.010 | 0.013 | 1 | 1 |

**Table 8  Best benchmarks results compared to ours.**

| Reference | Series | Model | MAPE |
|---|---|---|---|
| *Mallqui & Fernandes (2018)* | BTC | SVM:0.9-1(Relief) | 0.011 |
| *Patel et al. (2015)* | S&P BSE SENSEX | SVR | 0.009 |
| *Kazem et al. (2013)* | MSFT | SVR-CFA | 0.052 |
| | INTC | SVR-CFA | 0.045 |
| | NKSH | SVR-CFA | 0.046 |
| Our Work | BTC | LR | 0.007 |
| | ETH | MLR | 0.020 |
| | LTC | Univariate LSTM | 0.010 |
| | MSFT | MLR | 0.007 |
| | INTC | LR | 0.012 |
| | NKSH | LR | 0.009 |

For a direct feedback we report in Table 8 the best results obtained in the papers we compared to and our best ones. In the event that the best MAPE error results from different models, we consider the model whose computational complexity is the least as best. It is noticeable that our results outperform those obtained in the benchmark papers, providing notable contribution to the literature.

## CONCLUSIONS

The results, obtained considering the series in their totality, reflect the considerations made in the introduction of this paper. The predictions of the Bitcoin, Ethereum and Litecoin closing price series are worse, in terms of *MAPE* error, than those obtained for the benchmark series (Intel, Microsoft and National Bankshares). This is probably due to at least two reasons: high volatility of the prices and market immaturity for cryptocurrencies. This is confirmed by the statistics reported in Tables 1 and 2.

The results obtained partitioning the dataset into shorter sequences also confirmed the correctness of our hypothesis of identifying time regimes that do not resemble a random walk and that are easier to model, finding that best results are obtained using more than one previous price. It is worth noting that, with this novel approach, we obtained the best results for the Bitcoin price series rather than for the stock market series, as happened in the analysis of the series in their totality. As stated before, this is probably due to the high volatility of the Bitcoin price. In fact, it is no accident that the best result was found for the time regime identified by a translation step $h$ of 120, where the Bitcoin prices are more distributed around the mean, showing a lower variance. This is confirmed by the standard deviation values shown in Table 2.

It is important to emphasize that the innovative approach proposed in this paper, namely the identification of short-time regimes within the entire series, allowed us to obtain leading-edge results in the field of financial series forecasting.

Comparing our best result with those obtained in the considered benchmark papers, our result represents one of the best found in the literature. We highlight that we obtained, both for the Bitcoin and the traditional market series, better results than the benchmark ones. Precisely, for Bitcoin we obtained a *MAPE* error of 0,007, while the benchmark best one (*Mallqui & Fernandes, 2018*) is 0,011. For the stock market series our algorithms outperform those of benchmarks even more. In fact, our errors are as low as between 15% and 30% with respect to the reference errors reported in the literature.

Also for the Ethereum and Litecoin time series, the best results are those obtained with the time regimes approach, with a MAPE of 2% and 1% respectively.

As regards the implemented algorithms, the best results were found with both *regression models* and *LSTM* network. However, from the point of view of execution speed, the linear regression models outperform neural networks.

It is worth noting that, since Bitcoin and the other cryptocurrencies still are at an early stage, the length of the time series is limited, and future investigation might yield different results.

### Funding

This research is supported by the research project "EasyWallet" - POR FESR 2014-2020 - Asse 1, Azione 1.1.3 Strategia 2 "Creare opportunità di lavoro favorendo la competitività delle imprese" Programma di intervento 3 "Competitività delle imprese" Bando "Aiuti per progetti di ricerca e sviluppo" Principal Investigator: Michele Marchesi, and by the research project "Crypto-Trading"- POR FESR 2014-2020 - Asse 1, Azione 1.1.3 Strategia 2 "Creare opportunità di lavoro favorendo la competitività delle imprese". Programma di intervento 3 "Competitività delle imprese" Bando "Aiuti per progetti di ricerca e sviluppo": Roberto Tonelli. There was no additional external funding received for this study. The funders had no role in study design, data collection and analysis, decision to publish, or preparation of the manuscript.

### Grant Disclosures

The following grant information was disclosed by the authors:
EasyWallet:  POR FESR 2014-2020 - Asse 1, Azione 1.1.3 Strategia 2.
Creare opportunità di lavoro favorendo la competitività delle imprese.
Programma di intervento 3 "Competitività delle imprese" Bando "Aiuti per progetti di ricerca e sviluppo".
'Crypto-Trading"- POR FESR 2014-2020 - Asse 1, Azione 1.1.3 Strategia 2.
Creare opportunità di lavoro favorendo la competitività delle imprese.

### Competing Interests

The authors declare there are no competing interests.

### Author Contributions

- Nicola Uras performed the experiments, analyzed the data, performed the computation work, prepared figures and/or tables, and approved the final draft.
- Lodovica Marchesi analyzed the data, performed the computation work, prepared figures and/or tables, and approved the final draft.
- Michele Marchesi conceived and designed the experiments, analyzed the data, authored or reviewed drafts of the paper, and approved the final draft.
- Roberto Tonelli conceived and designed the experiments, analyzed the data, authored or reviewed drafts of the paper, and approved the final draft.

### Data Availability

The code and data are available in the Supplementary Files.

### Supplemental Information

Supplemental information for this article can be found online at http://dx.doi.org/10.7717/peerj-cs.279#supplemental-information.

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
