# Peer review of "Forecasting Bitcoin closing price series using linear regression and neural networks models"

_PeerJ Computer Science, doi:10.7717/peerj-cs.279_

## Round 0.1 · original submission · Major Revisions

Dear authors, Even though the two reviewers indicate minor reviews, the actual issues identified in there and suggestions recommended prompt me to recommend a major revision.

While you will need to address the reviewer's concerns point wise, you will have to pay particular attention in: (i) explaining your experimental set-up precisely, (ii) explaining how you explored the parameter space, and if only ad-hoc choices were made (as seem to be the case currently, e.g., reviewer 1 notes "200 time steps for each period. Why? Did they try different periods?"), then you have to rectify this by a more rigorous exploration of the same, (iii) demonstrate the efficacy of your approach with the study of at least one more (ideally a few more) cryptocurrencies, and finally, (iv) provide comparison (preferably quantitative) with recent related works.

·

Basic reporting

Review of the paper :
Forecasting Bitcoin closing price series using linear regression and neural networks models (#44798)

Sergio Focardi

1. BASIC REPORTING
Given the popularity gained by cryptocurrencies, forecasting Bitcoin prices is an important task. Therefore we consider this paper an important contribution.

However, the authors should address a number of points as follows
a. The paper includes rather naïve statements such as on line 165 when they write “To facilitate the understanding of this analysis we introduce the concept of unit root.” Unit root is a standard concept in econometrics, the authors should eliminate statements such as the one above. Simply mention techniques that are well known
b. Not clear why they compare forecasts of Bitcoin with those of three stocks. They should explain why they compare these forecasts
c. They divide the entire periods in partially overlapping subperiods. Now the idea that markets go through different regimes with different levels of predictability is has been exploited in many papers. However, how to determine the length of each subperiod is a critical issue, even more so given the limited length of series. They choose 200 time steps for each period. Why? Did they try different periods? This would be anticipation of information and it would weaken results. The authors should explain how they choose the length of the subperiods

Experimental design

2. EXPERIMENTAL DESIGN

They should explain how they choose the length of the subperiods. This is a critical issue that the authors should clarify.

Validity of the findings

3. VALIDITY OF THE FINDINGS

The findings are credible. Results are not very robust given the limited length of time series. However, this is a problem typical of any study of Bitcoin prices and should be accepted with the warning that future investigations might yield different results.

They should exhibit all results avoiding statements such as the one on line 364 “For the sake of brevity we only show the best results obtained on a specific time window….” Cherry picking past results is not a rigorous way of conducting econometric studies.

Additional comments

4. General comments
The paper is interesting but suffer from a certain lack of rigor. The authors should try to avoid useless explanations of well known methodologies and try to be precise about critical issues on testing

·

Basic reporting

In the experimental settings, I did not find any error; evaluation metrics are given and methods are described. Time regimes could be better formalized; for example a pseudocode could be given in addition to the current writing.

Experimental design

Experimental setting must be defined with more rigor. For example, instead of "We performed several experiments" the authors can tell what epoch and batch sizes were tried.

Validity of the findings

The results seem valid. The MAPE values are similar to what we have found in our Bitcoin price prediction experiments (2016 MAPE 0.003829 for the first 300 days). However note that this manuscript is a bit behind state of the art models that use graph metrics in addition to price and volume.

Additional comments

In this manuscript, the authors use past price and volume information to predict Bitcoin price. Their first attempt shows that there is no useful signal in the data; the price walks randomly. Next, the authors use shorter periods and make predictions for each period.

In addition to BTC, the authors predict three share prices. Using the shares gives a sense of how different BTC is. However, we also have other currencies, such as ether, litecoin and even tokens on platform blockchains, whose price could be predicted. The experiments should be expanded with at least one more cryptocurrency, ideally more. Data. with the same format, can be taken from coinmarketcap.

More comments below:


The abstract is a bit colloquial. Verbs such as "turn out to be" can be avoided.
30: It was possible to demosntrate can be given in active voice : ... we demonstrate...
43: please update the volume (from May 2019) with current values.
45: researcher names are missing in the reference entry, so they are shown as D. or R. in the text.
66: it turned out-> From this analysis we show that ...

Related work does not explain how these works are different from the proposed method. Specifically, how did they perform, what were the weaknesses in them? Some contrast sentences are needed. A major issue is that this manuscript does not consider related work that use onchain metrics other than the volume. For example an early work is "Using the bitcoin transaction graph to predict the price of bitcoin by A Greaves, B Au - No Data, 2015" (I have never met or communicated with these authors).

125: all extracted from ... Does this mean that the authors extracted themselves? Please consider using active voice to avoid such questions. It would be useful to give a URL for every website mentioned.

134: Bitcoin's place in total market capitalization is known as "Bitcoin dominance", currently over 60%. This number should be given with a citation such as https://coinmarketcap.com/charts/

164: references for DF tests and AC plots are complete books. At least, these should be cited with the page number that explains these concepts. Citing a whole book which is beyond a paywall is not helpful.

168: why having a unit root causes inference problems. The sentence needs to be expanded.

187 and 207 and 219: software without authors have blank names in citations, but these libraries have proper publications that you can cite (e.g., pandas: a foundational Python library for data analysis and statistics W McKinney - Python for High Performance and Scientific Computing, 2011).

250: every regime is 200 points wide-> is each point a day?

Table 4 and 6 results are surprising; between left and right panels, results differ by once cell only.

Conclusion: the first sentence can be shorter. What is meant by kindness of a method?

---

## Round 0.2 · accepted · Accept

The two reviewers had indicated minor revisions, and I had furthermore indicated additional tasks to be addressed. As per the review response, the authors have addressed most of these reasonably. Accordingly, I recommend accepting this revised version without further review.